# FEEDBACK-DRIVEN BEHAVIORAL SHAPING FOR SAFE OFFLINE RL

## ABSTRACT

Learning safe policies in offline reinforcement learning (RL) requires access to a cost function, but dense annotations are rarely available. In practice, experts typically provide only sparse supervision by truncating trajectories at the first unsafe action, leaving a single terminal cost label. We frame this challenge as a credit assignment problem: the agent must determine which earlier actions contributed to the violation to learn safer behavior. To address this, we propose an approach that redistributes sparse stop-feedback into dense per-step costs using return decomposition, and then integrates these inferred costs into constrained offline RL. Across highway driving and a simulated continuous control task, our method achieves substantially lower violation rates compared to baselines, while preserving reward performance.

## 1 INTRODUCTION

Safe reinforcement learning (RL) seeks to optimize performance while enforcing constraints on unsafe outcomes (Achiam et al., 2017; Gu et al., 2024). In offline RL, agents learn from fixed datasets without interacting with the environment, avoiding unsafe exploration but inheriting the safety profile of the behavior policy that generated the data (Fujimoto et al., 2019; Levine et al., 2020; Fujimoto & Gu, 2021). When that behavior policy is unaware of safety, naïve offline RL produces policies that replicate unsafe decisions. Incorporating safety constraints in this setting is particularly challenging, as costs are often not explicitly observed in the dataset.

A practical source of supervision arises when an expert provides trajectory-level "stop" feedback, where unsafe behavior is flagged by immediately halting execution of the trajectory. Each unsafe trajectory is therefore truncated at the first safety violation, yielding a binary cost signal of 1 at that step and 0 beforehand. This form of feedback is realistic in practice (for example, a human overseer or automated monitor terminating execution upon observing unsafe behavior) and has been proposed as a practical way to communicate safety constraints to an RL agent (Poletti, 2023). However, the resulting signal is extremely sparse, since only the final unsafe state is penalized, leaving earlier precursor decisions unaddressed. As in sparse reward problems in RL (Arjona-Medina et al., 2019), credit assignment becomes difficult: without further processing, the agent learns only to avoid the terminal unsafe state rather than anticipating earlier hazards.

In this work, we address the problem of offline safe reinforcement learning from trajectory-level stop feedback. We introduce the Redistribution-based Cost Inference (RCI) framework, an approach for converting sparse trajectory labels into dense per-step cost signals suitable for offline policy learning. As illustrated by Figure 1, RCI comprises three components: (i) an expert annotates unsafe trajectories with stop labels; (ii) a return decomposition algorithm redistributes these sparse labels into dense per-step costs by inferring which earlier actions contributed to unsafe outcomes; and (iii) an offline constrained RL algorithm trains a policy using the inferred costs. This approach enables agents to predict risks throughout trajectories rather than only learning from terminal feedback.

We compare RCI against several baselines on two domains: a highway driving task and a simulated robot control task, using datasets generated by unsafe, random, and mixed behaviour policies. Our results demonstrate that RCI significantly outperforms baselines, achieving substantially lower violation rates while maintaining comparable returns across both domains.

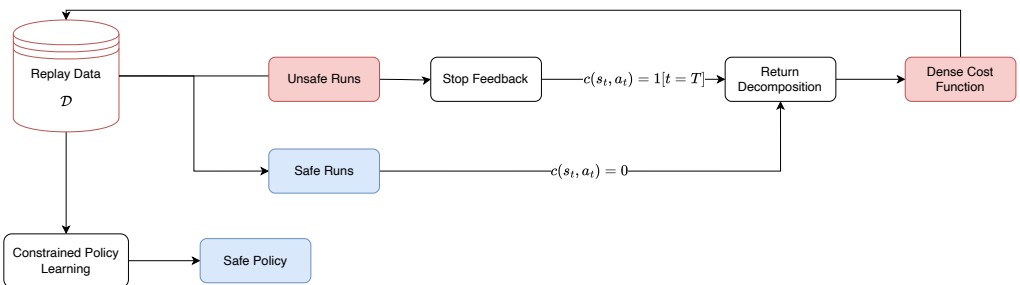

Figure 1: Overview of the Redistribution-based Cost Inference (RCI) framework. Unsafe trajectories are truncated at the first violation, producing sparse stop-feedback. A return decomposition model redistributes this terminal label into dense per-step costs, assigning higher penalties to precursor actions. These inferred costs, combined with rewards, form a dense cost-augmented dataset that a constrained offline RL algorithm uses to learn a safe policy.

## 2 PRELIMINARIES

In reinforcement learning (RL), a task is modeled as a Markov decision process (MDP), defined by the tuple $(\mathcal{S}, \mathcal{A}, P, r, \gamma)$. Here $\mathcal{S}$ is the state space, $\mathcal{A}$ is the action space, $P(s'|s, a)$ is the transition dynamics, $r(s, a)$ is the reward function, and $\gamma \in [0, 1)$ is the discount factor. A policy $\pi(a|s)$ generates trajectories $\tau = (s_0, a_0, s_1, \ldots, s_T)$ with cumulative reward $R(\tau) = \sum_{t=0}^{T} \gamma^t r(s_t, a_t)$ (Sutton et al., 1998). The objective is to find a policy that maximizes the expected return $\mathbb{E}[R(\tau)]$. This formulation captures the standard reinforcement learning setting where the agent seeks to maximize long-term reward.

In *offline* RL, the agent does not interact with the environment during training; instead, it learns from a fixed dataset $\mathcal{D} = \{\tau_i\}_{i=1}^{N}$ of trajectories generated by a behavior policy $\mu$. This avoids unsafe or costly trial-and-error, but introduces the problem of distributional shift: the learned policy $\pi$ may select actions outside the support of $\mu$, where value estimates are unreliable, a phenomenon known as extrapolation error. To address this, offline RL methods constrain the learned policy to remain close to the dataset distribution or penalize value estimates for unseen actions (Fujimoto et al., 2019; Kumar et al., 2020; Wu et al., 2019). For example, BCQ restricts the Bellman backup to dataset-supported actions, updating via

$$Q(s, a) \leftarrow r(s, a) + \gamma \max_{a' \in \mathcal{A}_\mathcal{D}(s')} Q(s', a'),$$

where $\mathcal{A}_D(s')$ denotes candidate actions generated to remain close to the dataset $\mathcal{D}$ (Fujimoto et al., 2019). These techniques help stabilize training and ensure the resulting policy remains within the regions supported by the data.

### 2.1 CONSTRAINED MARKOV DECISION PROCESSES

A constrained Markov decision process (CMDP) extends the MDP framework by introducing a cost function and a safety budget. Formally, a CMDP is given by $(\mathcal{S}, \mathcal{A}, P, r, c, \gamma, d)$, where $c(s, a)$ is a cost function representing constraint violations and $d$ is an allowable threshold. The agent's objective is to maximize $\mathbb{E}[R(\tau)]$ subject to $\mathbb{E}[C(\tau)] \leq d$, where $C(\tau) = \sum_{t=0}^{T} \gamma^t c(s_t, a_t)$ (Altman, 1998). This formulation balances performance and safety, for example, an autonomous vehicle should minimize accidents while still reaching its destination efficiently. A common approach to solving CMDPs is Lagrangian relaxation, where the problem is converted to maximizing $\mathbb{E}[R(\tau) - \lambda C(\tau)]$ with $\lambda$ adjusted until the cost constraint is satisfied (Achiam et al., 2017; Gu et al., 2024).

## 3 RELATED WORK

**Offline Safe Reinforcement Learning.** Offline safe RL combines the principles of CMDPs with offline training. The goal is to maximize expected return while ensuring that cost constraints are satisfied using only fixed datasets. Recent approaches incorporate conservative critics or cost penalties

into offline algorithms to enforce safety constraints Polosky et al. (2022); Xu et al. (2022). A key limitation is that most methods assume the cost function is explicitly available in the dataset.

**Cost Inference from Sparse Feedback.** Cost inference from sparse or delayed feedback represents a fundamental challenge in safe reinforcement learning, particularly when dense safety signals are unavailable. Traditional approaches assume access to well-specified per-step cost functions, but in practice, experts often provide only trajectory-level feedback or sparse annotations at critical unsafe transitions (Poletti, 2023; Low & Kumar, 2025).

Recent work has begun to address segment-level safety feedback. Low & Kumar (2025) propose TrACES, which learns a dense safety scoring model from sparse trajectory labels. Their approach is multiplicative and queries the labeler online for additional signals during training, enabling refinement of the inferred costs. Similarly, Chirra et al. (2024) introduce RLSF, which formulates surrogate objectives to transform segment-level feedback into classification tasks, but also relies on interactive access to the feedback in online learning. These settings differ fundamentally from ours: in offline RL the dataset is fixed, the labeler cannot be queried for more information, and any safety supervision must be incorporated post hoc without correction. Our framework addresses this offline constraint by redistributing trajectory-level labels into dense costs in a return-equivalent manner.

**Credit Assignment.** The core challenge for using sparse signal lies in **credit assignment**: determining which specific actions or state-action pairs within a trajectory contributed to eventual safety violations. This problem parallels sparse reward scenarios in standard RL, where agents must attribute delayed outcomes to earlier decisions (Arjona-Medina et al., 2019; Sutton et al., 1998). When safety feedback arrives only at trajectory termination points, naïve approaches that penalize solely the final unsafe state fail to capture the causal chain leading to violations.

Return decomposition techniques, originally developed for sparse reward problems (Arjona-Medina et al., 2019), provide another avenue for cost redistribution. Methods such as RUDDER train sequence models to predict cumulative returns and then redistribute terminal signals backward through time based on each step's contribution to the final outcome. While primarily designed for rewards, these techniques can be adapted to redistribute sparse cost signals, maintaining return equivalence while providing denser supervision for policy training (Zhang et al., 2023; Arjona-Medina et al., 2019).

## 4 METHODOLOGY

We formalize the problem of learning a safe policy from an offline dataset with sparse trajectory-level labels. The dataset $\mathcal{D} = \{\tau_i\}$ consists of trajectories $\tau = \{(s_0, a_0, r_0), (s_1, a_1, r_1), \ldots, (s_T, a_T, r_T)\}$. If a trajectory $\tau$ is deemed entirely safe, it has no termination label and runs until a normal end. If it is unsafe, the expert provides a stop label at the first unsafe transition $(s_t, a_t)$ and the trajectory is truncated at $t$. We can represent the expert's feedback as a function $\mathcal{F}(\tau)$ that returns a set of hazardous time indices in the trajectory (at most one per trajectory, the first unsafe index, or $\emptyset$ if none). Using this, we define a safety cost for each transition in the dataset as:

$$c(s_t, a_t) = \begin{cases} 1, & \text{if } t \in \mathcal{F}(\tau); \\ 0, & \text{otherwise.} \end{cases}$$

Each unsafe trajectory contributes a single nonzero cost at its termination point, and safe trajectories contribute zero costs throughout. This induces a sparse cost signal in $\mathcal{D}$. The goal is to learn a policy $\pi$ that maximizes the expected reward $\mathbb{E}_\pi[R(\tau)]$ while minimizing safety violations, ideally satisfying $\mathbb{E}_\pi[C(\tau)] \leq d$ for some desired cost budget $d$. The policy must be learned solely from $\mathcal{D}$ and the cost annotations, with no additional online interactions (García & Fernández, 2015; Achiam et al., 2017; Tessler et al., 2018).

The main difficulties are: (1) incomplete cost information, since we do not have a known cost function $c(s, a)$ for all states but only examples of failures in the dataset; (2) credit assignment, since costs are labeled at a single time step per unsafe trajectory and the agent must infer which earlier actions led to that outcome; and (3) distributional shift, since the learned policy may deviate from

the behavior that generated $\mathcal{D}$, leading to unknown states where safety is not guaranteed (Fujimoto et al., 2019; Kumar et al., 2020; Wu et al., 2019).

We address (1) and (2) by inferring a dense cost signal $\tilde{c}(s, a)$ from the dataset that assigns each transition a penalty, such that risky actions receive larger values. To address (3), we employ policy constraints based on BCQ (Fujimoto et al., 2019; Fujimoto & Gu, 2021) to keep the policy in regions covered by $\mathcal{D}$, under the assumption that $\mathcal{D}$ contains some safe behavior.

The first component of our approach is the feedback mechanism that supplies the initial safety labels in the data. In practice, having a human review every trajectory in a large dataset is costly, and in realistic deployments one might use a combination of automated detectors and human oversight. A stop label at $(s_t, a_t)$ is interpreted as a cost $c(s_t, a_t) = 1$. To ensure the agent does not learn from unsafe parts of trajectories, we discard any experience after the stop point. This yields a processed dataset $\mathcal{D}_{\text{safe}}$ of transitions with each transition labeled safe or unsafe, with unsafe transitions being rare.

To address the credit assignment problem, we introduce a return-decomposition approach to infer a dense cost representation from the sparse labels. We use the RUDDER algorithm, which trains a predictive sequence model on each trajectory to estimate the remaining cumulative cost from each state-action pair (Arjona-Medina et al., 2019).

Formally, let each trajectory $\tau = (s_0, a_0, \ldots, s_T, a_T)$ be labeled by the expert with an episodic cost $C(\tau) \in \{0, 1\}$, where $C(\tau) = 1$ if the trajectory was truncated at the first unsafe transition and $C(\tau) = 0$ otherwise. We train a sequence model $\hat{C}(s_{0:t})$ to predict the total episodic cost $C(\tau)$ given the trajectory prefix up to time $t$. The model is optimized by minimizing the prediction error of $\hat{C}(s_{0:T})$ with respect to the sparse label $C(\tau)$. Once trained, the redistributed per-step cost is defined by the difference in predicted return between successive prefixes: $\tilde{c}_t = \hat{C}(s_{0:t}) - \hat{C}(s_{0:t-1})$, with the baseline cost $\hat{C}(s_{0:-1}) = 0$ by convention. This decomposition ensures return equivalence: $\sum_{t=0}^{T} \tilde{c}_t = C(\tau)$ (Arjona-Medina et al., 2019), as detailed in Appendix A.1.

Intuitively, if the predicted cumulative cost rises sharply at a particular timestep, the corresponding state-action is assigned higher $\tilde{c}_t$, reflecting its contribution to eventual failure. Conversely, at the failure point itself, the raw sparse label may be redistributed backward, assigning more "blame" to precursors.

These $\tilde{c}_t$ values serve as dense, trajectory-consistent cost signals that replace the sparse labels. While in this work we instantiate redistribution using RUDDER (Arjona-Medina et al., 2019), any return-equivalent decomposition methods such as GRD (Zhang et al., 2023) could be used, making RCI decomposition agnostic.

### 4.1 POLICY OPTIMIZATION

It is worth noting that our framework is modular. In principle, one can pair cost redistribution step can be paired with any offline RL algorithm. Similarly, while we use RUDDER for decomposition and redistribution, any alternative method that is return equivalent and assigns meaningful per-step signals can be substituted. Our approach is outlined in Algorithm 1.

We use BCQ-Lag from the OSRL library (Liu et al., 2023) to perform constrained offline learning using the redistributed cost signals. BCQ consists of a Q-network $Q_\theta(s, a)$, a variational autoencoder $G_\omega(s)$ to model the behavior policy, and a perturbation network $\xi_\phi(s, a)$ that refines sampled actions. At each step, candidate actions are sampled via $a = G_\omega(s) + \xi_\phi(s, a)$ and evaluated by the Q-function.

To enforce safety, the Q-update incorporates a cost penalty with a Lagrange multiplier $\lambda$. The target value becomes:

$$y = r(s, a) + \gamma \max_{a' \in \mathcal{A}_{\text{cand}}(s')} \left[ Q_{\theta'}(s', a') - \lambda \tilde{c}(s', a') \right], \tag{1}$$

where $\mathcal{A}_{\text{cand}}(s')$ are actions sampled from $G_\omega(s')$ and perturbed by $\xi_\phi$. The Q-networks are trained to minimize the squared Bellman error with this target (Fujimoto et al., 2019). The policy networks

$(G_\omega, \xi_\phi)$ are updated to maximize $Q(s, a)$, promoting high-reward, low-cost actions (Liu et al., 2023).

The Lagrange multiplier is updated after each batch using:

$$\lambda \leftarrow \max\left(0, \lambda + \alpha(C_{\text{batch}} - d)\right), \tag{2}$$

where $C_{\text{batch}}$ is the average cost in the sampled batch and $d$ is the cost budget. This dynamic adjustment penalizes the policy when safety violations exceed the threshold (Altman, 1998).

---

**Algorithm 1** Redistribution-based Cost Inference (RCI) Framework

---

1: **Input:** Dataset $\mathcal{D} = \{\tau_i\}_{i=1}^N$, Labeler $\mathcal{L}$, Return decomposition algorithm $\mathcal{A}_{\text{decomp}}$, Constrained Offline RL algorithm $\mathcal{A}_{\text{OSRL}}$, Cost budget $d$
2: **Output:** Safe policy $\pi$
3: **for** each trajectory $\tau_i \in \mathcal{D}$ **do**
4:    $F(\tau_i) \leftarrow \mathcal{L}(\tau_i)$                                        ▷ Feedback Collection
5:    $C(\tau_i) \leftarrow \begin{cases} 1 & \text{if } F(\tau_i) \neq \emptyset \\ 0 & \text{otherwise} \end{cases}$
6:    **for** $t = 0, 1, \ldots, T_i$ **do**
7:       $c^{\text{sparse}}(s_t, a_t) \leftarrow \begin{cases} 1 & \text{if } t \in F(\tau_i) \\ 0 & \text{otherwise} \end{cases}$
8:    **end for**
9: **end for**
10: $\hat{C} \leftarrow$ Train sequence model with $\mathcal{A}_{\text{decomp}}$ on $\{(\tau_i, C(\tau_i))\}$       ▷ Return Decomposition
11: **for** each trajectory $\tau_i \in \mathcal{D}$ **do**
12:    **for** $t = 0, 1, \ldots, T_i$ **do**
13:       $\tilde{c}_t \leftarrow \hat{C}(s_{0:t}) - \hat{C}(s_{0:t-1})$                      ▷ with $\hat{C}(s_{0:-1}) = 0$
14:    **end for**
15: **end for**
16: $\mathcal{D}_{\text{dense}} \leftarrow \{(s_t, a_t, r_t, \tilde{c}_t)\}$                       ▷ Constrained Policy Learning
17: $\pi \leftarrow \mathcal{A}_{\text{OSRL}}(\mathcal{D}_{\text{dense}}, d)$
18: **return** $\pi$

---

## 5 EXPERIMENTS

**Environments.** We evaluate our approach on two benchmark environments: `HighwayEnv` (Leurent, 2018) and `Safe-FetchReach` (de Lazcano et al., 2024). In `HighwayEnv`, the ego vehicle must navigate highway traffic while avoiding collisions. The state space includes vehicle positions, velocities, and surrounding traffic configurations, while actions control acceleration and throttle. `Safe-FetchReach` is a robotic manipulation task in which a 7-DOF robotic arm must reach target positions while avoiding a spherical hazard region. The state space encompasses joint angles, gripper position, and target coordinates, with actions controlling joint velocities. Full specifications of state, action, reward, and unsafe labeling are provided in Appendix A.2.

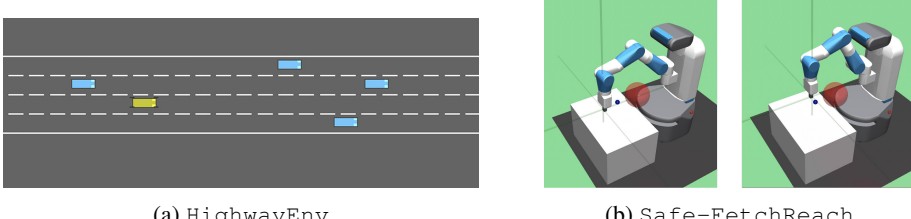

(a) `HighwayEnv`                                    (b) `Safe-FetchReach`

Figure 2: Snapshots of the benchmark environments. (2a) an ego vehicle navigating highway traffic. (2b) a 7-DOF robotic arm reaching a target while avoiding a hazard region.

**Dataset Generation.** For both environments, we generate offline datasets using PPO-trained behavioral policies that prioritize task performance while disregarding safety constraints (Schulman et al., 2017). In `HighwayEnv`, the behavioral policy optimizes for speed, progress, and lane changes, resulting in aggressive driving behaviors including close overtaking and occasional collisions with other vehicles. The policy explicitly prioritizes the fast lane and does not maintain safe following distances. In `Safe-FetchReach`, the behavioral policy focuses solely on reaching the goal position, completely ignoring the unsafe spherical region during trajectory execution. It should be noted that Figure 4 and 8 show the evaluations on policy trained on the mixed data composition, where offline data consists of rollouts from PPO and random policy.

**Feedback Collection.** To simulate realistic safety feedback, we use an automated evaluator to examine each trajectory and provide binary stop/continue labels. The evaluator identifies the first unsafe transition in each trajectory and discards all subsequent steps, mimicking real-world expert intervention. For `HighwayEnv`, unsafe transitions are detected when the ego vehicle approaches within a critical distance threshold of other vehicles. For `Safe-FetchReach`, violations occur when the gripper enters the predefined hazard region. This labeling protocol results in sparse safety signals with terminal costs at violation points.

**Baselines and Implementation.** We compare against three baselines: (1) Reward-Only (Vanilla), which ignores safety costs entirely; (2) Sparse, which uses the raw terminal cost labels without redistribution; and (3) Hazard, which trains a classifier to assign $c(s_t, a_t) = P_1(s_t, a_t) + P_2(s_t, a_t)$, where $P_1$ is the probability that the $(s_t, a_t)$ is in an unsafe trajectory and $P_2$ is the probability that $(s_{t+1}, a_{t+1})$ is the unsafe event. All methods use identical BCQ architectures with actor-critic networks, VAE for action generation, and perturbation networks for policy regularization (Fujimoto et al., 2019). For experiments, RCI integrates RUDDER-based cost redistribution (3b), where we instantiate an LSTM as the sequence model for return decomposition and prediction residuals are folded at the end of trajectories to ensure return equivalence (Arjona-Medina et al., 2019). Complete hyperparameter settings for all domains are provided in Appendix 1.

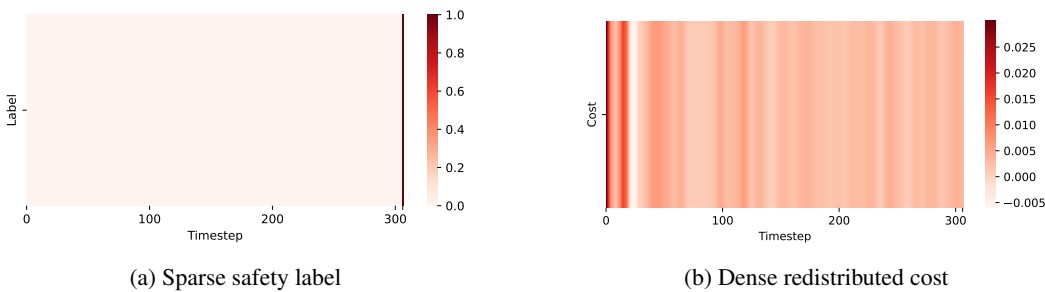

(a) Sparse safety label

(b) Dense redistributed cost

Figure 3: Cost inference on a sampled trajectory from `HighwayEnv`. (3a) Sparse supervision yields only a terminal cost at the unsafe step. (3b) RUDDER redistributes this signal into dense per-step costs, assigning higher values to precursor actions.

**Evaluation Protocol.** For each environment and baseline, we perform hyperparameter selection over cost budget values entirely in the offline setting. Specifically, we conduct an empirical sweep using the percentile range $P_q \mid q \in [10, 50]$ of the dataset's trajectory cost distribution, where $P_q$ denotes the $q^{th}$ percentile. All methods are trained exclusively on the offline datasets, with three independent runs per configuration. Final policies are then evaluated over 1000 online episodes. The best policy is chosen based on achieving the lowest violation rate while maintaining competitive task reward performance, reflecting the trade-off between safety and reward optimization. Importantly, the same safety criterion is applied for offline data labeling and online evaluation.

## 5.1 RESULTS

We present all experiments and ablations on the `HighwayEnv`. We then include a complementary evaluation on `Safe-FetchReach`, to demonstrate that our method generalizes continuous control tasks, though we restrict analysis to the core results.

In Figure 4, we report two key metrics: **Return** (normalized task reward) and **Violation Rate** (percentage of episodes with safety violations). Normalized return is computed per task as $R_{\text{norm}} = \frac{R - R_{\min}}{R_{\max} - R_{\min}}$, where $R_{\min}$ and $R_{\max}$ are taken from the pooled distribution of episode returns across all evaluated policies and seeds on that task Liu et al. (2023). This ensures values are comparable across baselines within the same task.

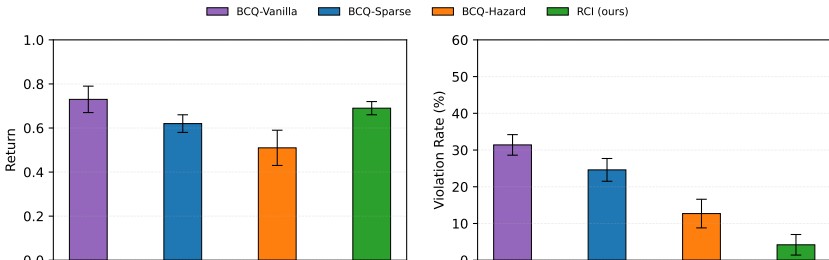

Figure 4: Performance comparison on `HighwayEnv`. The plot shows normalized return and violation rate, highlighting the safety–performance trade-off across methods. Higher return is better, lower violation rate is better. Bars show mean over 3 seeds; error bars denote standard error.

To evaluate the robustness of our approach under realistic conditions, we conduct two additional ablations on `HighwayEnv` by evaluating performance across different data compositions and imperfect supervision.

**Dataset Composition.** In Figure 5, we assess RCI's robustness across different data collection regimes, we evaluate performance on datasets generated by three distinct behavioral policies with fundamentally different exploration and decision-making patterns: (i) *PPO* datasets generated by task-optimized policies that exhibit goal-directed behavior with focused exploration around high-reward regions; (ii) *Random* datasets collected via uniform random action sampling; and (iii) *Mixed* datasets combining rollouts from both PPO and Random policies, with an equal number of episodes drawn from each policy to create heterogeneous data distributions that reflect realistic scenarios where multiple data sources contribute to offline datasets. This experiment examines whether cost redistribution mechanism remains effective across varying behavioral policy characteristics.

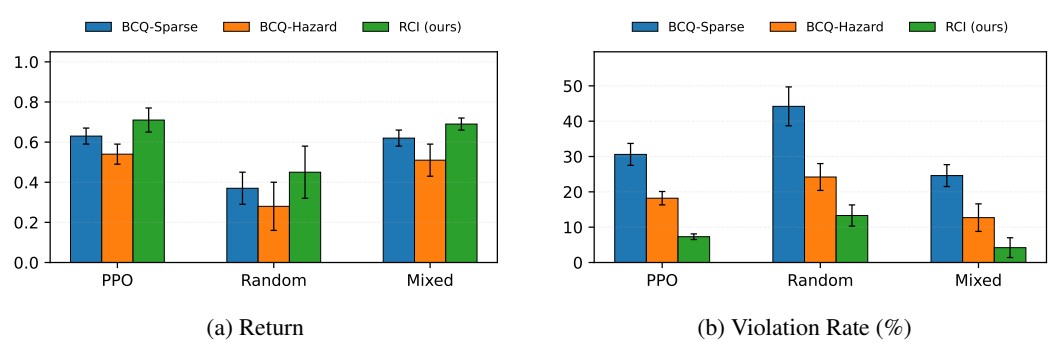

(a) Return  (b) Violation Rate (%)

Figure 5: Effect of dataset composition on `HighwayEnv`. Bars show mean over three seeds, error bars denote standard error. Each policy is evaluated for 1,000 episodes per seed using ground-truth violations. We use the same safety budget and model-selection rule across methods.

**Label Noise.** We simulate imperfect annotations by randomly shifting the termination index up to 15 steps earlier or later than the true unsafe step $t^*$, while ensuring the perturbed index remains within the trajectory length $T$. This bounded perturbation shifts the termination label earlier (false positive) or later (false negative) relative to the actual unsafe transition, simulating realistic annotation errors where a human or an automated labeler have limited precision in identifying the exact moment of safety violation. Safe trajectories without violations remain unchanged. We fix the cost budget $d$ constant to isolates the causal effect of label noise on reported metrics.

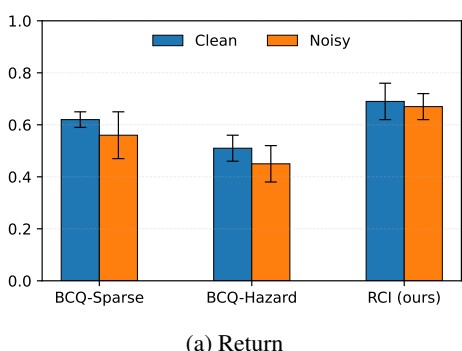

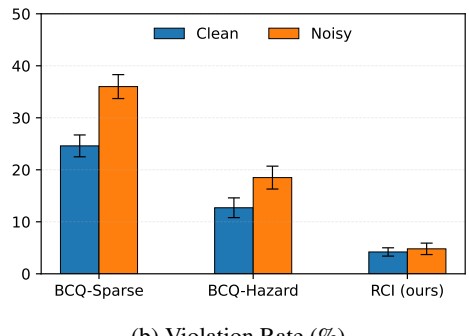

(a) Return

(b) Violation Rate (%)

Figure 6: Label-noise experiment on `HighwayEnv` with cost budget $d$ fixed to the value selected in the core results 4. Error bars show standard error across 3 seeds; evaluation uses 1000 episodes and ground-truth safety events.

Noise in termination time alters the constraint signal but not the logged rewards, as a result, return remains stable (6a) while violation rates on baselines respond more sharply to misaligned safety supervision (6b).

To complement the quantitative results, Figure 7 illustrates RCI's learned policies under different safety budgets in `HighwayEnv`. With a restrictive budget, the policy interprets the trajectory-level cost signal conservatively, exiting the highway entirely to avoid potential violations. Under a balanced budget, the implicit cost specification instead permits progress along the road while still maintaining safe distances from surrounding vehicles. These qualitative rollouts highlight how trajectory-level supervision, when redistributed through RCI, shapes distinct safety–performance trade-offs.

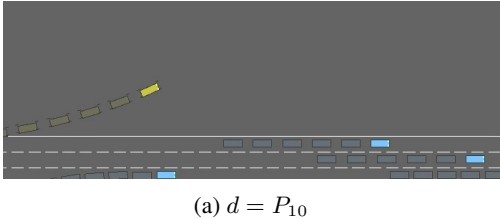

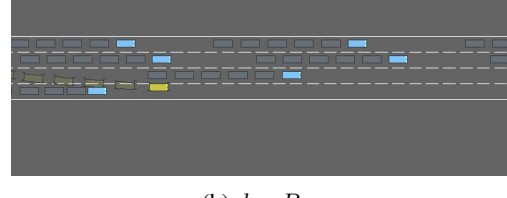

(a) $d = P_{10}$

(b) $d = P_{30}$

Figure 7: Trajectories from RCI's learned policy in `HighwayEnv` under restrictive (7a) and balanced (7b) safety budgets $d$, where $P$ denotes the $q^{th}$ percentile of dataset's episodic cost distribution

### 5.1.1 CONTINUOUS CONTROL TASK

To complement our extensive evaluation on `HighwayEnv`, we report core results on the `Safe-FetchReach` domain under the same offline RL protocol (data generation, safety budgets, and evaluation metrics) described in Section 5. This task involves a 7-DOF robotic arm that must reach a sampled Cartesian target while avoiding entry into a predefined spherical hazard region (see Appendix A.2 for full environment details). The purpose here is to test whether RCI extends to a distinct continuous-control setting with geometric safety constraints.

We observe the same trend as in `HighwayEnv`: RCI produces policies that achieve competitive task rewards while substantially reducing violation rates compared to baselines. The redistributed per-step costs assign penalties not only at the moment of hazard entry but also to precursor actions that increase risk, enabling the policy to anticipate and avoid unsafe regions rather than reacting only at failure points.

The results in Figure 8 demonstrate that RCI generalizes beyond driving tasks and remains effective in high-dimensional robotic control. By shaping policies according to redistributed costs, RCI adapts flexibly to new safety specifications while preserving performance.

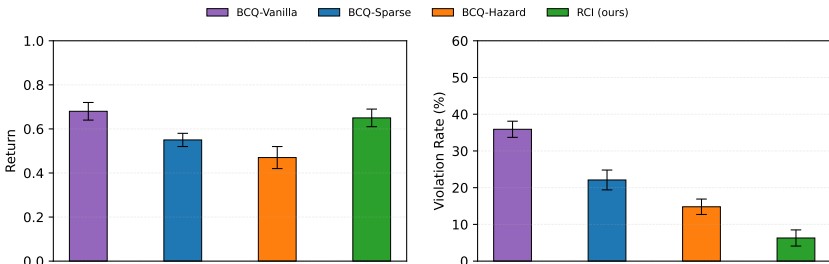

Figure 8: Performance comparison on `Safe-FetchReach`. The plot shows normalized return and violation rate, highlighting the safety–performance trade-off across methods. Higher return is better, lower violation rate is better. Bars show mean over 3 seeds; error bars denote standard error.

To provide further intuition, Figure 9 shows vector fields of the learned policies in `Safe-FetchReach` under different safety budgets. With a balanced budget, the implicit cost specification allows the policy to pursue the target while steering away from the hazard region. Under a restrictive budget, the inferred costs dominate, causing the policy to bias strongly away from the hazard even if it prevents reaching the goal. This visualization highlights how trajectory-level supervision, redistributed through RCI, induces distinct control strategies in high-dimensional continuous tasks.

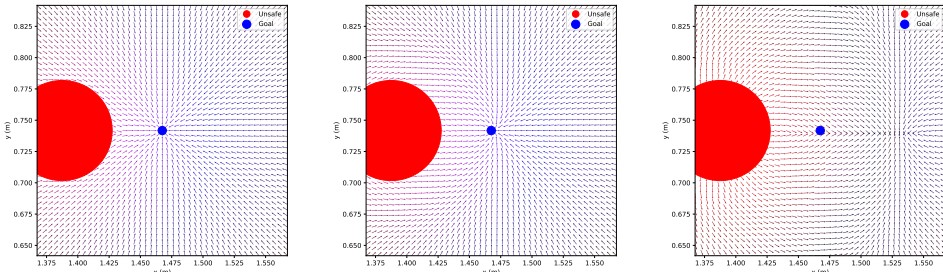

Figure 9: Vector fields of RCI's learned policies in `Safe-FetchReach` across three safety budgets from the empirical percentile sweep. Arrows depict policy actions, and color encodes the critic's value estimates. From left to right, safety budgets decrease: higher budgets permit riskier trajectories, intermediate budgets balance hazard avoidance with goal-reaching, and the lowest budgets enforce strong avoidance of the hazard even at the expense of task success.

## 6 DISCUSSION AND FUTURE WORK

A central limitation is the reliance on the quality of trajectory-level labels. If the supervision is noisy or biased, redistribution faithfully preserves this signal at the trajectory level but also spreads the existing errors across steps. While the label noise ablation shows robustness to small amounts of label noise, systematic or adversarial errors in feedback remain a challenge. In addition, we assume datasets provide sufficient coverage of safe behaviors to guide policy constraints. When unsafe trajectories dominate or safe coverage is sparse, redistribution may not capture actionable safety signals.

Taken together, our findings highlight how return redistribution transforms sparse trajectory-level stop signals into dense costs that make safe offline learning feasible. By preserving the supervision signal while shaping per-step behavior, RCI consistently achieves lower violation rates without sacrificing task performance. This demonstrates that effective safety in offline RL does not require dense labels, but rather careful credit assignment of sparse ones.

Looking ahead, the implications extend beyond the benchmarks studied here. As datasets in robotics, driving, and healthcare increasingly contain incomplete or noisy safety information, methods like RCI offer a path to leverage them without requiring costly dense annotation.

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

## A APPENDIX

### A.1 SIGNAL PRESERVATION

A key intuition behind our framework is that, in the offline setting, supervision must be preserved since no additional feedback can be queried post hoc. For safety constraints, this means that dense penalties must remain equivalent to the sparse labels so that the constraint budget reflects the same information available in the dataset.

Following Arjona-Medina et al. (2019), we define redistributed per-step costs as

$$\tilde{c}_t = \hat{C}(s_{0:t}) - \hat{C}(s_{0:t-1}) + \delta_t, \qquad \hat{C}(s_{0:-1}) = 0, \tag{3}$$

where $\hat{C}(s_{0:t})$ is the predicted cumulative cost from a trajectory prefix and $\delta_t$ is a compensation term ensuring return equivalence.

Summing across the trajectory yields a telescoping series:

$$\sum_{t=0}^{T} \tilde{c}_t = \sum_{t=0}^{T} \left[ \hat{C}(s_{0:t}) - \hat{C}(s_{0:t-1}) + \delta_t \right] \tag{4}$$

$$= \hat{C}(s_{0:T}) + \sum_{t=0}^{T} \delta_t. \tag{5}$$

Since $\hat{C}$ is trained to approximate the expert-provided label $C(\tau) \in \{0, 1\}$, the compensation term accounts for the prediction error, ensuring that

$$\sum_{t=0}^{T} \tilde{c}_t = C(\tau). \tag{6}$$

This telescoping property guarantees that redistribution preserves the sparse signal in aggregate Arjona-Medina et al. (2019). If expert labels are noiseless, the redistributed costs equal the true episodic cost. If there is label error, the redistribution inherits it but does not introduce further distortion. In offline RL, where the dataset is fixed, this preservation property ensures that the agent is no worse than the quality of the supervision itself.

### A.2 ENVIRONMENT DETAILS

#### A.2.1 HIGHWAYENV

**Task.** The agent controls an ego vehicle on a multilane highway populated with traffic, aiming to maintain forward progress while avoiding unsafe maneuvers.

**State space.** Each state is a fixed-size array of kinematic features for the ego and $V - 1$ nearby vehicles, including presence, relative position $(x, y)$, velocities $(v_x, v_y)$, and orientation $(\cos h, \sin h)$. Features are normalized and expressed in ego-centric coordinates, with zero-padding to maintain fixed dimensionality.

**Action space.** Actions are two-dimensional continuous controls $a = [a_{\text{throttle}}, \delta_{\text{steer}}]^\top \in [-1, 1]^2$, mapped to acceleration $\dot{v} \in [-5, 5]$ m/s$^2$ and steering $\delta \in [-0.785, 0.785]$ rad.

**Reward function.** We modify the native reward to emphasize acceleration, simulating a safety-unaware ego vehicle: $R(s, a) = \alpha \cdot \frac{v - v_{\min}}{v_{\max} - v_{\min}} + \gamma \cdot \dot{v} - \beta \cdot \mathbf{1}\{\text{collision}\}$, with $\alpha, \beta, \gamma > 0$. This shaping encourages forward velocity and aggressive acceleration, sometimes leading to collisions in the absence of explicit safety constraints.

**Unsafe criterion.** A transition is labeled unsafe when the ego vehicle comes within distance $0.2$ of another vehicle:

$$c_{sparse}(s, a) = \mathbf{1}\Big\{ \min_{j \in \{1, \ldots, V-1\}} \|p_{\text{ego}} - p_j\|_2 \leq 0.2 \Big\}.$$

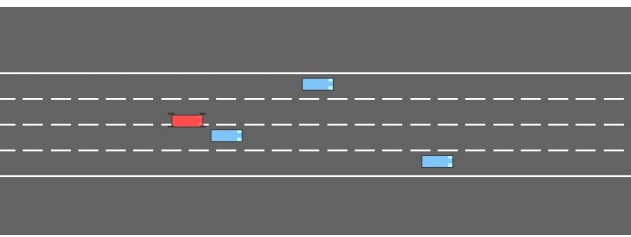

Figure 10: `HighwayEnv` unsafe criterion: a transition is labeled unsafe when the ego vehicle comes within distance $0.2$ of another vehicle.

### A.2.2 SAFE-FETCHREACH

**Task.** A 7-DOF Fetch robot must move its end-effector to a sampled Cartesian target. We introduce a hazard region to assess safety.

**State space.** Observations are tuples $(o, g_{\text{ach}}, g_{\text{des}})$, where $o \in \mathbb{R}^{10}$ encodes end-effector position, finger joint states, and velocities, and $g_{\text{ach}}, g_{\text{des}} \in \mathbb{R}^3$ denote achieved and desired goals.

**Action space.** Actions are $a = [\Delta x, \Delta y, \Delta z, a_{\text{grip}}]^\top \in [-1, 1]^4$, with the first three mapped to Cartesian displacements and the fourth controlling the gripper (unused here).

**Reward function.** The dense shaping reward is $R(s, a) = -\|g_{\text{ach}} - g_{\text{des}}\|_2$.

**Unsafe criterion.** We define a spherical hazard region $\mathcal{H} = \{p \in \mathbb{R}^3 : \|p - h\|_2 \leq r\}$ with center $h$ and radius $r$. Unsafe labels are assigned at the first transition where $g_{\text{ach}} \in \mathcal{H}$:

$$c_{sparse}(s, a) = \mathbf{1}\{g_{\text{ach}} \in \mathcal{H}\}.$$

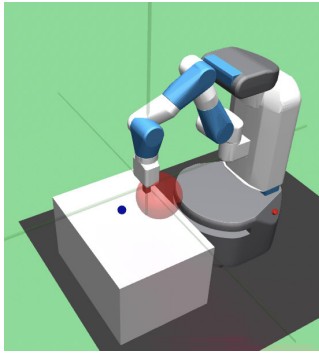

Figure 11: `Safe-FetchReach` unsafe criterion: a transition is labeled unsafe when the end-effector enters the spherical hazard region $\mathcal{H}$.

## A.3 IMPLEMENTATION DETAILS

Table 1: Hyperparameters for BCQ-Lagrangian across benchmark environments

| Hyperparameter | HighwayEnv | Safe-FetchReach |
|---|---|---|
| *Network Architecture* | | |
| Actor Hidden Sizes | [256, 256] | [256, 256] |
| Critic Hidden Sizes | [256, 256] | [256, 256] |
| VAE Hidden Sizes | [750, 750] | [750, 750] |
| Number of Q-networks | 2 | 2 |
| Number of Q-cost networks | 2 | 2 |
| *Learning Rates* | | |
| Actor Learning Rate | 1e-4 | 1e-4 |
| Critic Learning Rate | 3e-4 | 3e-4 |
| VAE Learning Rate | 1e-3 | 1e-3 |
| *Algorithm Parameters* | | |
| Discount Factor ($\gamma$) | 0.99 | 0.99 |
| Soft Update Rate ($\tau$) | 0.005 | 0.005 |
| Perturbation Scale ($\lambda$) | 0.75 | 0.75 |
| VAE Beta ($\beta$) | 0.5 | 0.5 |
| Threshold Parameter ($\phi$) | 0.05 | 0.05 |
| Sample Action Number | 10 | 10 |
| *Training Configuration* | | |
| Batch Size | 256 | 128 |
| Update Steps | 5e5 | 1e5 |
| Evaluation Episodes | 10 | 5 |
| Evaluation Frequency | 5,000 | 1,000 |

Table 2: Hyperparameters for PPO used for data generation

| Hyperparameter | HighwayEnv | Safe-FetchReach |
|---|---|---|
| *Network Architecture* | | |
| Policy Hidden Sizes (pi) | [256, 256] | [256, 256] |
| Value Hidden Sizes (vf) | [256, 256] | [256, 256] |
| *Learning Rates* | | |
| Learning Rate | 5e-4 | 3e-4 |
| *Algorithm Parameters* | | |
| Discount Factor ($\gamma$) | 0.80 | 0.98 |
| GAE $\lambda$ | 0.95 | 0.95 |
| Clip Range | 0.20 | 0.20 |
| Entropy Coefficient | 0.0 | 0.0 |
| Value Function Coef | 0.5 | 0.5 |
| *Training Configuration* | | |
| n_steps | 1,024 | 1,024 |
| Batch Size | 256 | 256 |
| Epochs per Update | 10 | 10 |
| Total Timesteps | 500,000 | 300,000 |
| Evaluation Episodes | 10 | 10 |
| Evaluation Frequency | 10,000 | 10,000 |
| Early-Stop Success Threshold | - | 0.95 |

**Usage of Large Language Models.** We used a large language model (LLM) to polish the writing of this paper. No other parts of the research, including the design, implementation, experiments, or analysis, involved the use of LLMs.

