# OpenReview forum: "Feedback-driven Behavioral Shaping for Safe Offline RL"
_ICLR.cc/2026/Conference — ICLR 2026 Conference Withdrawn Submission_

### Official Review · Reviewer_ugML · 2025-10-25

**Soundness:** 2
**Presentation:** 3
**Contribution:** 2
**Rating:** 2
**Confidence:** 4

**Summary:**

This paper tackles the problem of safe offline reinforcement learning (RL) when explicit constraint-violation cost signals are unavailable. Instead, the authors leverage termination signals (episodes that end upon reaching an unsafe state) as sparse feedback. The proposed method addresses the credit assignment problem caused by such sparse signals via cost return decomposition (similar to RUDDER) and mitigates distributional shift using BCQ-Lagrangian.

Overall, the paper presents an intuitive and relevant direction for making offline RL safer and more practical under limited supervision. However, the experimental evaluation and comparison scope are currently too limited. With a more comprehensive set of environments, datasets, and baselines, this work could develop into a meaningful contribution to the safe RL community.

**Strengths:**

1. The paper considers a practical and underexplored setting where safety feedback is sparse and trajectory-level only.
2. The cost redistribution approach eliminates the need for dense manual labeling by experts or LLMs, providing a scalable alternative.
3. The manuscript is well-written and logically organized, making it easy to follow the technical contributions.
4. The notion of “stop-feedback” is interesting, as it aligns with real-world safety data collection, where unsafe trajectories are truncated early for safety reasons.

**Weaknesses:**

1. Limited evaluation scope: The paper evaluates only two domains (highway driving and a single continuous-control task). This limits generalizability and makes it difficult to assess robustness.
2. Narrow set of baselines: Only BCQ-Lag and a few internal variants are considered. I suggest that the authors include several recent safe offline RL methods (e.g., FISOR [1], LSPC [2], CCAC [3], etc., from recent ICLR editions) for a fair and up-to-date comparison.
3. Potentially biased redistribution: The cost redistribution appears to allocate high penalties to early steps (as seen in Fig. 5), which may not reflect true causal responsibility for safety violations.
4. Lack of diagnostic analysis: There is limited analysis verifying whether redistributed costs correlate with actual risk-contributing states or actions.
5. Assumption on return decomposition: The approach assumes that a trajectory-level model can meaningfully infer stepwise costs, but this assumption is neither empirically validated nor theoretically supported.
6. Minor Suggestion - CMDP formulation (Sec. 2.1): The paper describes the objective as “minimizing accidents while reaching the destination efficiently.” Would it not be more natural to express this as maximizing the probability of reaching the destination while constraining the probability of accidents below a threshold? This distinction might better capture the constrained optimization nature of CMDPs.



[1] Zheng, et al. "Safe offline reinforcement learning with feasibility-guided diffusion model." arXiv preprint arXiv:2401.10700 (2024).

[2] Koirala, et al. "Latent Safety-Constrained Policy Approach for Safe Offline Reinforcement Learning." arXiv preprint arXiv:2412.08794 (2024).

[3] Guo, et al. "Constraint-conditioned actor-critic for offline safe reinforcement learning." 2025.

**Questions:**

1. Discarding unsafe trajectories (Line 172): The paper mentions discarding experience after the stop point. Is learning from unsafe parts of trajectories necessarily detrimental? In safe RL, unsafe transitions often help train a more accurate cost critic. Could their exclusion bias the learning process?
2. Return decomposition reliability: The framework assumes that an autoregressive model trained on total trajectory cost can infer meaningful per-step costs. Could it degenerate into focusing only on the terminal event (assigning near-zero cost to all previous steps)? How can such behavior be prevented or detected?
3. Hazard baseline ambiguity: The description of the Hazard baseline is unclear. Both ​P1 and P2 seem to depend on the current state–action pair, but the text suggests otherwise. Could the authors clarify whether this is a typographical or conceptual inconsistency?
4. Fig. 5: Why is the redistributed cost high and concentrated at the beginning of episodes? This could suggest misplaced credit assignment, where early states are overly “blamed” for eventual failures. Is this an artifact of the RUDDER-style decomposition or a property of the dataset?
5. Negative redistributed costs: Figure 5 shows some negative redistributed costs, which contradicts the typical CMDP definition where costs are nonnegative. What do these negative values represent, and are they preserved during policy training? What are their implications for constraint satisfaction?

---

### Official Review · Reviewer_uYfz · 2025-10-31

**Soundness:** 3
**Presentation:** 3
**Contribution:** 2
**Rating:** 4
**Confidence:** 3

**Summary:**

This paper proposes RCI (Redistribution-based Cost Inference), a framework for learning safe policies in offline RL when only sparse trajectory-level stop feedback is available. The key challenge is credit assignment: trajectories are truncated at the first unsafe action, yielding a single terminal cost label, but the agent must determine which earlier actions contributed to the violation. RCI addresses this through three components: (1) expert annotates unsafe trajectories with stop labels, (2) return decomposition (RUDDER) redistributes sparse terminal costs into dense per-step costs by training an LSTM to predict cumulative cost and computing differences between successive trajectory prefixes, and (3) constrained offline RL (BCQ-Lag) trains a safe policy using the inferred costs. Experiments on HighwayEnv and Safe-FetchReach demonstrate that RCI achieves substantially lower violation rates compared to baselines (Reward-Only, Sparse, Hazard) while maintaining competitive task returns.

**Strengths:**

- Trajectory-level stop feedback is realistic and mirrors real-world safety supervision (human oversight, automated monitors)
- Return decomposition and policy learning are decoupled—any return-equivalent method can replace RUDDER, and any offline RL algorithm can replace BCQ-Lag
- Well-motivated problem, intuitive visualization of cost redistribution (Figure 3), and qualitative policy analysis (Figures 7, 9)

**Weaknesses:**

- Only two environments (highway driving, 7-DOF robot) with relatively simple safety constraints (distance thresholds, spherical hazards)—unclear how this scales to complex tasks
- TrACES and RLSF are discussed in related work but not compared experimentally; authors claim offline setting differs but adaptation should be possible
- Requires dataset to contain sufficient safe behavior for BCQ constraints; performance when unsafe trajectories dominate is not analyzed
- Only tested with BCQ-Lag; generalizability to other offline safe RL methods (CPQ, COptiDICE) not demonstrated

**Questions:**

1. How does RCI compare to these segment-level feedback methods in the offline setting? Could you adapt them by removing online queries?

2.  What happens when unsafe trajectories comprise >80% of the dataset? Does the method degrade gracefully or fail catastrophically?

3.  How does RUDDER's LSTM handle trajectories with 500+ timesteps? Do prediction errors accumulate and degrade cost redistribution quality?

4. Does RCI work with other offline safe RL algorithms beyond BCQ-Lag? How does it pair with model-based methods or diffusion policies?

---

### Official Review · Reviewer_8CQb · 2025-11-01

**Soundness:** 2
**Presentation:** 2
**Contribution:** 2
**Rating:** 2
**Confidence:** 5

**Summary:**

This paper presents a method for Safe RL, particularly as it relates to stop conditions related to safety violations and the sparse feedback signals that come as a result of such a setting. The paper leverages notions of return decomposition and related notions from credit assignment, to create a dense surrogate signal that allows for prediction of risk before a violation and throughout a trajectory/episode rather than relying only on this terminal violation.

**Strengths:**

The manuscript presents a clearly written and accessible introduction: the motivation of addressing sparse safety/violation feedback in offline reinforcement learning is nicely articulated, and the high‑level approach of redistributing trajectory‑level cost signals into per‑step cost signals is easy to follow. This clarity is commendable.

**Weaknesses:**

There are several concerns regarding the positioning of the work, the depth of preliminary exposition, the novelty/impact of the contribution, and methodological clarity. See the questions below.

**Questions:**

1. Inadequate Related Work & Positioning
The manuscript mentions some prior work on credit assignment / return decomposition in RL, yet it fails to provide a systematic comparison of how those methods match or differ from the present work. For example, RUDDER (Arjona‑Medina et al., 2019) explicitly introduces return decomposition via contribution analysis to handle delayed rewards. The current submission should clearly locate itself relative to this: in what respects is cost‑redistribution simply a straightforward application of the same idea, and in what respects is it novel?
Similarly, there is a body of work on hindsight credit assignment (e.g., attributing which action or event in a trajectory caused an eventual outcome) and on temporal‑difference (TD) approaches to propagate costs or rewards backward. The manuscript mentions these only superficially, without contrasting the advantages/disadvantages of each branch (e.g., TD‑based propagation vs decomposition via sequence modeling) and then highlighting the unique benefit of the proposed approach.
In the safe/offline RL domain, the manuscript cites only two benchmark datasets/environments. Yet recent work on offline safe‑RL benchmarks (e.g., Liu et al., 2023)https://arxiv.org/pdf/2306.09303  provides broader suites across many tasks. The manuscript should discuss those works and compare how the present experimental choices relate to them. In summary: the authors need to systematically summarise the branches of prior work (reward/return redistribution, hindsight credit assignment, TD/back‑propagation of cost, offline safe RL benchmarks), contrast their approach to each (assumptions, algorithmic choices, strengths/weaknesses), and then clearly articulate the novelty of their contribution.


2. Preliminaries Section is Too Text‑Book and Low Value
The manuscript devotes a large portion of the “Preliminaries” section to textbook‑level background (standard MDP/RL definitions, Bellman equations). One suggestion given the limited space available is that this section should be concise, with most of the background moved to the appendix. The main text should focus on how the present work modifies the standard setup and why those modifications matter for cost/violation signals in offline RL.
Recommendation: shorten the background (or move it to the appendix) and allocate space to emphasise the new modeling/algorithmic choices of this paper.

3. Contribution / Significance Appears Limited
The claimed contribution—applying return‑decomposition / credit‑assignment to sparse cost signals and then training a constrained offline policy—reads as a relatively straightforward adaptation of existing ideas rather than a fundamental advance. For a top‑tier venue, please articulate: (i) any new theoretical insights/guarantees about cost redistribution; (ii) specific modeling/algorithmic challenges unique to costs versus rewards; and (iii) empirical evidence on widely used benchmarks showing a clear advantage over prior art.
The manuscript should explicitly discuss how it differs from directly applying RUDDER‑style decomposition to costs. If the only difference is that prior work focused on reward assignment while this work focuses on cost assignment, it is unlikely to meet the novelty bar without further technical contributions.

4. Methodological/Modeling Clarifications and Suggestions
Per‑step cost is defined via the incremental change in predicted episodic cost between successive prefixes.  That is, $\hat{C}(s_{0:-1})$ and $\tilde{c}(s_{t}) = \hat{C}(s_{0:t})-\hat{C}(s_{0:t-1})$.
This attribution assumes that the change in violation probability at time $t$ is the cost for that step. While intuitive, it may under‑represent interactions among factors across time. Consider discussing or comparing with an explicit conditional model of violation risk, e.g., modeling $P(v | s_{0:t-1}, a_t)$ (where $v$ is a violation) and attributing contributions accordingly.

5. Additional Limitation: Dataset Exhaustiveness
One important assumption underlying the proposed method is that the offline dataset is sufficiently exhaustive—that is, it covers all or most of the unsafe cases that the policy might encounter. If the dataset fails to include a wide enough range of unsafe scenarios, the cost‑redistribution model will be unable to learn accurate risk attribution, resulting in poor generalization to unseen unsafe situations. In practice, this means the method’s reliability heavily depends on the coverage breadth of the dataset; limited datasets may lead to unrecognized safety violations or over‑optimistic policy behavior.

---

### Official Review · Reviewer_wo3e · 2025-11-01

**Soundness:** 2
**Presentation:** 2
**Contribution:** 1
**Rating:** 2
**Confidence:** 4

**Summary:**

The paper studies offline safe RL with sparse "stop-at-first-violation" feedback. It trains a sequence model to predict episodic cost from this offline feedback dataset and obtains per-step costs via return decomposition (RUDDER-style). These inferred dense costs are then plugged into a constrained offline RL algorithm (e.g., BCQ-Lag) to train a safer policy.

**Strengths:**

- **Practical problem**: implicit safety is a real-world challenge in safe RL.
- **Feedback structure**: explicitly models stop feedback at the first unsafe transition (though this raises positioning concerns (hard constraint vs. soft CMDP; see Weakness 1).
- **Clear pipeline**: clear framework from dense cost prediction to applying them in standard constrained offline RL.

**Weaknesses:**

## Paper Positioning

1. **Labeling protocol and CMDP objective mismatch**: Dataset is issued a stop label at the first unsafe transition. This seems to be closer to hard constraint RL which demands zero visitation to unsafe states. CMDP formulation in 2.1 indicates a soft constraint where a soft budget is imposed, i.e., a policy is considered safe if cost incurred is below a certain user-specified budget. This makes the paper's intended setting unclear.

2. **Implicit budget contradiction**: In CMDP, budget is user specified. The paper relabels safety cost to {0, 1} but chooses budget $d$ from the offline dataset percentiles. It's unclear whether this percentile refers to the unknown ground-truth cost or the relabeled proxy, and why practitioners should set $d$ from dataset percentiles when $d$ is supposed to be implicit and unobserved / unknown. By contrast, Chirra et al. (2024) assumes budget is known while Low & Kumar (2025) infers budget from safety data.

3. **State-only cost modeling**: Line 180-187 describes how dense cost is obtained in the paper. The cost predictor only depends on state sequences $\hat C(s_{0:t})$ despite earlier $C(\tau)$ with $\tau$ defined as the whole state action sequences $\tau = (s_0, a_0, ..., s_T, a_T)$. In the problem formulation (Section 2.1), the ground-truth cost is also defined as dependent on both state and actions, i.e. $c(s_t, a_t)$. This creates a mismatch between problem formulation and model.

## Methodology

4. **Cascading residual errors**: In Appendix A.1, it is claimed that the sum of compensation terms $\sum_{t=0}^T \delta_t$ maps to prediction error and is dropped from Eq6. But each $\delta_t$ depends on the prediction for smaller subsegments. With sparse labels, the smaller subsegment has no direct supervisory ground-truth signal and thus the compensation term at each timestep could be high. Consequently, the sum of all the inferred cost may not approximate or close to the expert-provided label, challenging Eq6's equality in practice.

5. **Inferred cost significance**: Return decomposition techniques don't guarantee that decomposed costs identify true causal precursors of safety events. For example, Fig 3b shows very early timesteps (close to step 0) with high cost and it's challenging to see how the initial steps near $t \approx 0$ caused an unsafe event 300+ steps later. The redistributed cost could be highly arbitrary than truly pointing to safety-significant steps.

6. **Dense or sparse redistribution**: The return decomposition techniques also don’t guarantee that the redistribution is dense. The inferred cost could still collapse to Dirac-delta-like attribution since the provided label only happens at one timestep.

7. **Length bias**: The proposed framework truncates unsafe trajectories and redistributes costs to earlier steps which approximately sums to $C(\tau)$. Since $\sum_{t=0}^T \tilde{c}_t$ approximates $C(\tau)$, shorter unsafe trajectories could inflate per-step costs relative to longer ones (or the untruncated safe trajectories). It's useful to analyze this bias further.

8. **Limited novelty**: The proposed method is said to be modular: it uses prior return decomposition methods to redistribute sparse cost label and uses standard constrained RL as algorithm. As written, this reads as an application of known redistribution + plug-and-play with constrained BCQ, with little new algorithmic substance. Furthermore, the return decomposition methods seem to be generally applicable to RL problems (and RL algorithm-agnostic), it's unclear why the paper positions it solely within the offline safe RL setting.

9. **Computation overhead**: To query the dense cost at each timestep, it requires $T+1$ classifier calls for each trajectory. It would be helpful to discuss the scalability to long-horizon problems.

## Evaluation

10. **Limited baseline choices**: The evaluation doesn't seem sufficient given the choice of baselines. They are all restricted to the same BCQ method with varying knowledge of the cost. The reward-only and sparse baselines are weak baselines, since it's expected that cost redistribution would outperform those two. I'd think that an Oracle baseline that has near-complete knowledge of the safety cost and other safety cost inference baseline method should be included.

11. **Puzzling choice of hazard baseline**: Hazard baseline should be described in more detail. The P_1 classifier outputs probability of a state-action appearing in unsafe trajectory. This looks highly similar to RLSF (Chirra et al., 2024). This could be a baseline on its own. It is unclear why it is mixed with an equally weighted P_2 as a hazard baseline.

12. **Under-justified P_2 classifier as baseline**: P_2 classifier is not well defined. It predicts whether a state-action tuple is the unsafe event. Given the sparse labeling of such event, the classifier should be heavily biased towards classifying as safe (due to extreme class imbalance). The paper does not explain its utility or the rationale for mixing with P_1.

13. **Evaluation scope**: Only two tasks are evaluated, each with custom safety criteria, e.g. (i) within critical distance threshold of other vehicles and (ii) presence in predefined hazard region. These tasks and safety definition are not the standard safe RL benchmark / metrics. Furthermore, it necessitates creation of an entirely new in-house offline dataset collected by custom collection policy. This creates new challenges in assessing the validity of the results and raises questions on whether the result is transferable to another offline dataset.

14. **Uninformative noisy label ablation**: The noise model shifts the unsafe event earlier or later. Shifting unsafe event earlier would make the policy more conservative and avoid getting close to unsafe region altogether. Shifting it later would make it more adventurous. Aggregating to population level, the conservative-adventurous effect would largely cancel out. Realistic mislabeling (e.g. unsafe trajectory classified as safe and vice versa) should be tested instead.

15. **Metric misalignment**: Evaluation reports violation rate without the expected cost, deviating from standard safe RL benchmarks. The budget selection procedure (using dataset percentiles) is not aligned with explicit budget, implicit budget nor hard-constraint setting. All of these create challenges in assessing the validity of results.

## Reproducibility

16. The paper makes use of in-house offline datasets collected by its own custom collection policies. None of the dataset, model params, or source code is made available to support reproducibility assessment.

**Questions:**

## Positioning

1. Is the trajectory labeled with a hard stop at the first instance of unsafe transition? Or is a soft budget considered during annotation? The problem formulation is a soft CMDP, does the labeling strategy align with that?

2. If the budget is implicit / unknown, how is choosing $d$ from dataset percentiles justified at deployment? This seems to require online peeking to infer an appropriate value of $d$ at deployment.

## Method

3. Why is action excluded in the sequence model $\hat{C}(s_{0:t})$ if the goal is to learn $C(\tau)$?

4. Is the sum of inferred costs, $\sum_{t=0}^T \tilde{c}_t$, consistently close to the expert-provided label? Are there cases where it falls below 0 or exceeds 1?

## Experiments

5. Is there any study performed to check whether high $\tilde{c}_t$ steps correspond to safety-critical steps?

---

### Note · Authors · 2025-11-17

**Comment:**

Thank you to the reviewers and area chair for the feedback. I have decided to withdraw the submission in order to address the concerns raised. I appreciate the input and will use it to strengthen future work.

**Withdrawal Confirmation:**

I have read and agree with the venue's withdrawal policy on behalf of myself and my co-authors.